# A Novel Decentralized Game-Theoretic Adaptive Traffic Signal Controller: Large-Scale Testing

**DOI:** 10.3390/s19102282

**Published:** 2019-05-17

**Authors:** Hossam M. Abdelghaffar, Hesham A. Rakha

**Affiliations:** 1Department of Computers & Control Systems, Engineering Faculty, Mansoura University, Mansoura, Dakahlia 35516, Egypt; hossamvt@vt.edu; 2Center for Sustainable Mobility, Virginia Tech Transportation Institute, Virginia Tech, Blacksburg, VA 24061, USA; 3Charles E. Via, Jr. Dept. of Civil and Environmental Engineering, Director of the Center of Sustainable Mobility, Virginia Tech Transportation Institute, Virginia Tech, Blacksburg, VA 24061, USA

**Keywords:** traffic signal control, game theory, decentralized control, large-scale network control

## Abstract

This paper presents a novel de-centralized flexible phasing scheme, cycle-free, adaptive traffic signal controller using a Nash bargaining game-theoretic framework. The Nash bargaining algorithm optimizes the traffic signal timings at each signalized intersection by modeling each phase as a player in a game, where players cooperate to reach a mutually agreeable outcome. The controller is implemented and tested in the INTEGRATION microscopic traffic assignment and simulation software, comparing its performance to that of a traditional decentralized adaptive cycle length and phase split traffic signal controller and a centralized fully-coordinated adaptive phase split, cycle length, and offset optimization controller. The comparisons are conducted in the town of Blacksburg, Virginia (38 traffic signalized intersections) and in downtown Los Angeles, California (457 signalized intersections). The results for the downtown Blacksburg evaluation show significant network-wide efficiency improvements. Specifically, there is a 23.6% reduction in travel time, a 37.6% reduction in queue lengths, and a 10.4% reduction in ${\mathrm{CO}}_{2}$ emissions relative to traditional adaptive traffic signal controllers. In addition, the testing on the downtown Los Angeles network produces a 35.1% reduction in travel time on the intersection approaches, a 54.7% reduction in queue lengths, and a 10% reduction in ${\mathrm{CO}}_{2}$ emissions compared to traditional adaptive traffic signal controllers. The results demonstrate significant potential benefits of using the proposed controller over other state-of-the-art centralized and de-centralized adaptive traffic signal controllers on large-scale networks both during uncongested and congested conditions.

## 1. Introduction

Traffic growth and limited available capacity within the roadway system produces problems and challenges for transportation agencies. Traffic congestion affects traveler mobility and has an impact on air quality, and consequently on public health. The stopping and starting in traffic jams burns fuel at a higher rate than the smooth rate of travel, and contributes to the amount of emissions released by vehicles that create air pollution and are related to global warming [1]. Reduction in traffic congestion improves traveler mobility and accessibility, while also reducing vehicle fuel consumption and emissions.

Traffic congestion in 2013 cost Americans $124.2 billion [2], and this number is projected to rise to $186.2 billion in 2030. Traffic signal controllers attempt to optimize various traffic variables (e.g., delay, queue length, and energy and emission levels), by optimizing signal control variables, including the cycle length, the phasing scheme and sequence, the phase split, and the offset. Most of the currently implemented traffic signal systems can be categorized into one of the following categories: fixed-time control (FP), actuated control (ACT), responsive control, or adaptive control [3].

An FP control system is developed off-line using historical traffic data to compute traffic signal timings; real-time traffic data is not taken into account, and the duration and order of all phases stay fixed without any adaptation to real-time traffic demand fluctuations [4]. Previous studies have found this approach to only be appropriate for under-saturated conditions and traffic flows that are stable or relatively stable [5]. By comparison, ACT systems respond to changes in traffic demand patterns by communicating with the controller based on the presence or absence of vehicles as identified by local detectors installed at intersection approach stop lines. While ACT has been proven to generally perform better than FP for very low demand levels, it still offers no real-time optimization to adapt to traffic fluctuations, and may result in long network queues [6]. Adaptive systems have the potential to alleviate traffic congestion by adjusting signal timing parameters in response to real-time traffic fluctuations. These systems use detector inputs, historical trends, and predictive models to predict vehicle arrivals at intersections, and then use the predictions to determine the best gradual changes in cycle length, phase splits, and offsets to minimize vehicle delays or queue lengths [7]. Some examples in this category are: the Split Cycle Offset Optimization Tool (SCOOT) [8], a macroscopic model that minimizes delay and the number of vehicle stops at all intersection approaches, and performs effectively in under-saturated traffic conditions. The Sydney Coordinated Adaptive Traffic System (SCATS) [9] operates in a centralized hierarchical mode, and allocates green times to the phases of greatest need. OPAC [10] optimizes an objective function for a specified rolling horizon using dynamic-programming-based traffic prediction models that require a traffic environment state transition probability model, which can be difficult to generate. TR2 and UTCS-1 [11], optimized off-line, are incapable of handling stochastic variations in traffic patterns.

The operation of actuated and adaptive controllers is constrained by minimum and maximum cycle lengths, green indication durations and offsets, and also require going through a pre-defined sequence of phases. In addition, some systems use hierarchies that either partially or totally centralize decisions, rendering them more susceptible to failures. Hierarchies make scaling these systems up more difficult, relatively more complex to operate, and more expensive [13].

Various computational intelligence-based techniques have been investigated in the domain of traffic signal optimization domain, and are still under continuous research and development, using fuzzy sets, genetic algorithms, reinforcement learning, and neural networks. Genetic algorithms compute the optimal solution using an evolutionary process of possible solutions [13,14]; it solves simple networks and deals with static traffic volumes. However, as the network increases in size, the search space involved in finding effective signal plans increases significantly, and a large amount of centralized computing power is required. Pappis [15] proposed the first signal controller using fuzzy logic for an isolated intersection. Ella [16] proposed a neuro-fuzzy controller, where the parameters of the fuzzy membership functions were adjusted using a neural network. The neural learning algorithm in Ella’s work was reinforcement learning, which was found to be successful at constant traffic volumes, but failed when the traffic demand changed rapidly. The choice of the membership functions (building blocks of fuzzy set theory) are important for a particular problem since they affect a fuzzy inference system. As a traffic control system is a complex large-scale system with many interactive factors, it is more appropriate to use fuzzy control for isolated intersections [17].

Several approaches have been proposed for designing traffic signal controllers using neural networks [18,19]. Most of these works are based on a distributed approach, where an agent is assigned to update the traffic signals of a single intersection. Neural networks also adapt very slowly to changing traffic parameters, where on-line learning has to take place continuously. Some networks require multiple models to be maintained for various times within a day. Most intelligence-based approaches are still being researched and are thus under development or have only been implemented and tested on an isolated intersection, so their effectiveness for controlling a large-scale traffic network is also unknown.

Reinforcement learning is inspired by behavioral psychology [20]. It is a machine learning approach which allows agents to interact with the environment, attempting to learn the optimal behavior based on the feedback received from interactions. The feedback may be available right after the action, or several time steps later, which makes the learning more challenging [21]. Abdulhai et al. [22] applied a model-free Q-learning technique to a simple two-phase isolated traffic signal in a two-dimensional road network. Salkham et al. [23] applied a Q-learning strategy that allowed an agent to exchange rewards with its neighbors on 64 signalized intersections. The state-action space was simple and very time coarse. Each agent decided the phase splits every two cycles, which did not capture of the rapid dynamics of congestion–coordination between the agents actions was missing. Studies have considered the use of RL algorithms for traffic control, but they are very limited in terms of network complexity and traffic loadings, so that realistic scenarios, over saturated conditions, and transitions from under saturation to over saturation (and vice versa) have not been fully explored.

Game theory studies the interactive cooperation between intelligent rational decision makers with the specific goal of cooperating and benefiting from reaching a mutually agreeable outcome. It has been widely used in economic, military, communication applications [24,25], model traveler route choice behavior [26], control connected vehicle movements [27], and to in-route guidance [28]. The literature indicates that investigation of game-theoretic traffic signal control is very limited. Bargaining theory is related to cooperative games through the concept of Nash bargaining (NB). A bargaining situation is defined as a situation in which multiple players with specific objectives cooperate and benefit by reaching a mutually agreeable outcome [29]. The bargaining process is the procedure that bargainers follow to reach an agreement (outcome) [30], and the bargaining outcome is the result of the bargaining process [31,32].

Traffic flow is affected by a number of factors, including weather, time-of-day, day-of-week, and unpredictable events, such as special events, incidents, and work zones. Consequently, traffic control strategies could be improved if control systems responded not only to actual conditions, but also adapted their actions to transient conditions. Due to the stochastic nature of traffic flows, an adaptive control strategy that adjusts to stochastic changes is needed. Cycle-free strategies may present an innovative and less restrictive means of accommodating variations in traffic conditions.

Traffic signal controllers can be categorized as centralized or decentralized. Centralized systems require a reliable and direct communication network between a central computer and the local controllers. The main advantage of these systems is that they allow for traffic signal coordination. However, decentralized systems offer many advantages over centralized control systems as they are computationally less demanding and require only relevant information from adjacent intersections/controllers. Robustness is also guaranteed in decentralized control systems, because if one or more controllers fail, the remaining controllers can take over some of their tasks. Decentralized systems are scalable and easy to expand by inserting new controllers into the system. Additionally, decentralized systems are often inexpensive to establish and operate, as there is no essential need for a reliable and direct communication network between a central computer and the local controllers in the field.

To mitigate traffic congestion, a novel de-centralized traffic signal controller, considering a flexible phasing sequence and cycle-free operation, using a NB game-theoretic framework (DNB) is developed. The proposed controller was implemented and evaluated in the INTEGRATION microscopic traffic assignment and simulation software [33,34,35]. INTEGRATION is a microsopic model that replicates vehicle longitudinal motion using the Rakha–Pasumarthy–Adjerid collision-free car-following model, also known as the RPA model [36]. The RPA model captures vehicle steady-state car-following behavior using the Van Aerde model [37,38]. Movement from one steady state to another is constrained by a vehicle dynamics model described in [39,40]. Vehicle lateral motion is modeled using lane-changing models described in [35]. The model estimates of vehicle delay were validated in [41], while vehicle stop estimation procedures are described and validated in [42]. Vehicle fuel consumption and emissions are modeled using the VT-Micro model [43,44,45]. The developed controller was compared to the operation of a decentralized phase split and cycle length controller (PSC) [6], and a fully coordinated adaptive phase split-cycle length and offset optimization controller (PSCO) to evaluate its performance, where PSCO is based on the REALTRAN (REAL-time TRANsyt) controller that emulates the SCOOT system [46,47]. The DNB controller was implemented and evaluated on large-scale networks consisting of 38 (Blacksburg) and 457 (downtown Los Angeles) signalized intersections.

This paper describes the application and the testing of the proposed DNB controller on large-scale networks and is organized as follows. Section 2 describes the developed de-centralized traffic signal controller using a game-theoretic framework. Section 3 presents the experimental setup and results of a large-scale study in the town of Blacksburg, Virginia, consisting of 38 signalized intersections. Section 4 describes the experimental setup and the experimental results of a large-scale study on a downtown network in Los Angeles, California, consisting of 457 signalized intersections. Section 5 presents a summary and conclusions drawn from these studies.

## 2. Traffic Signal Controller

This section describes the NB solution for two players (Section 2.1), Section 2.2 describes how the NB approach is adapted and extended to control a multi-phase (player) signalized intersection (DNB), and Section 2.3 describes the de-centralized mechanism of the DNB controller over an entire transportation network.

### 2.1. NB Solution for Two Players

A bargaining situation is defined as a situation in which multiple players with specific objectives cooperate and benefit by reaching a mutually agreeable outcome (agreement). In bargaining theory, there are two concepts: the bargaining process and the bargaining outcome.

The bargaining process is the procedure that bargainers follow to reach an agreement (outcome). Nash adopted an axiomatic approach that abstracts the bargaining process and considers only the bargaining outcome [31]. The bargaining problem consists of three basic elements: players, strategies, and utilities (rewards). Bargaining between two players is illustrated in the bi-matrix shown in Table 1. Each player, namely $P_{1}$ and $P_{2}$, has a set of possible actions $A_{1}$ and $A_{2}$, whose outcome preferences are given by the utility functions *u* and *v*, respectively, as they take relevant actions.

The space (*S*) shown in Figure 1, is the set of all possible utilities that the two players can achieve; the vertices of the area are the utilities where each player chooses their pure strategy. The disagreement or the threat point $d=(d_{1},d_{2})$ corresponds to the minimum utilities that the players want to achieve. The threat point is a benchmark, and its selection affects the bargaining solution. Each player attempts to choose their threat point in order to maximize their bargaining position. Subsequently, a bargaining problem is defined as the pair (*S*,*d*) where $S\in R^{2}$ and d∈S such that *S* is a convex and compact set, and there exists some s∈S such that s>d.

Nash’s theorem states that there exist a unique solution satisfying four axioms (Pareto efficiency, symmetry, invariance to equivalent utility representation, and independence of irrelevant alternatives), and this solution is the pair of utilities $\left(u^{*},v^{*}\right)$ that solves the following optimization problem:(1)$$\begin{array}{c} \;\;\max_{u,v}\;\left(u-d_{1}\right)\left(v-d_{2}\right)\\ s.t.\left(u,v\right)\in S,\left(u,v\right)\geq \left(d_{1},d_{2}\right) \end{array}$$

The NB solution $\left(u^{*},v^{*}\right)$ of this optimization problem can be calculated as the point in the bargaining set that maximizes the product of the players utility gains relative to a fixed threat point.

### 2.2. DNB Traffic Signal Controller for Multi-Players

This section describes the game model and the DNB solution for multi-players (N), and shows how the model is adapted (from Section 2.1) and applied to control a multi-phase signalized intersection. First, a four-phase scheme for a four-legged intersection is used, assuming four players (N=4), to represent the intersection phases as shown in Figure 2, with protected, leading main street left-turn phases.

In the game model, the four phases are modeled as four players $P_{1}$, $P_{2}$, $P_{3}$, and $P_{4}$ in a four-player cooperative game. For each player (phase), there are two possible actions: maintain $\left(A_{1}\right)$ or change $\left(A_{2}\right)$. These actions produce the state for the traffic signal. Specifically, the action maintain maintains the traffic signal (i.e., if it is displaying a green indication, it will remain green; if it is displaying a red indication, it will remain red). The action change entails changing the state of the traffic signal (i.e., if it is displaying a green indication, it will switch its state by first introducing a yellow indication followed by a red indication; if it is red, it will switch to a green indication) in the simulated time interval. The combinations of phases offer four possibilities, where only one player holds the green indication and all others hold red indications [48].

The INTEGRATION software is a microscopic traffic simulation model that traces individual vehicle movements every deci-second. Driver characteristics such as reaction times, acceleration and deceleration levels, desired speeds, and lane-changing behavior are examples of stochastic variables that are modeled in INTEGRATION. The threshold speed is fixed and assigned to the entire network (chosen to be equal to the typical pedestrian speed, $s^{Th}$= 4.5(km/h)). We continuously check the vehicle speeds when they are within the threat distance from the approach stop bar. If the vehicle (*v*) speed ($s_{v}^{t}$) is less than the threshold speed ($s^{Th}$) at time (*t*), the vehicle is assigned to the queue, and the current queue length associated with the corresponding lane (*l*) is updated. Once the vehicle’s speed exceeds ($s^{Th}$) the queue length is updated (i.e., shortened by the number of vehicles leaving the queue). This is formulated mathematically as
(2)$$q_{l}^{t}=\sum_{v\in v_{l}^{t}}q_{v}^{t}$$
(3)$$q_{v}^{t}=\left\{\begin{array}{cc} \;\;1 & \mathrm{if}\;s_{v}^{t-1}>s^{Th}\;\&s_{v}^{t}\leq s^{Th}\\ -1 & \mathrm{if}\;s_{v}^{t-1}\leq s^{Th}\;\&s_{v}^{t}>s^{Th}\\ \;\;0 & \left\{\begin{array}{c} \mathrm{if}\;s_{v}^{t-1}\leq s^{Th}\;\&s_{v}^{t}\leq s^{Th}\\ \mathrm{if}\;s_{v}^{t-1}>s^{Th}\;\&s_{v}^{t}>s^{Th} \end{array}\right. \end{array}\right.$$
$q_{l}^{t}$ is the number of queued vehicles in lane *l* at time *t*. The index (t−1) is used to refer to the previous time step. In this case the previous deci-second as the INTEGRATION model tracks vehicle movements at a frequency of 10 hertz.

The utilities (rewards) for each player (phase) in the game can be defined as the estimated sum of the queue lengths in each phase after applying a specific action. The estimated queue length after applying a specific action is calculated according to the following equation:(4)$$Q_{P}\left(t+\Delta t\right)=\sum_{l\in P}q_{l}^{t}+Q_{inl}\Delta t-Q_{outl}\Delta t$$
where Δt is the updating time interval, $q_{l}^{t}$ is the current queue length at time *t*, $Q_{P}\left(t+\Delta t\right)$ is the estimated queue length after Δt for phase *P*, $Q_{inl}$ is the arrival flow rate (veh/h/lane), and $Q_{outl}$ is the departure flow rate (veh/h/lane).

The NB solution is extended to four players (N=4) with a four-dimensional utility space and threat points. The solution for the four-phase NB problem can be formulated as:(5)$$\begin{array}{c} \;\;\;\max_{\left(u_{1},\ldots ,u_{N}\right)}\prod_{i=1}^{N}\left(u_{i}-d_{i}\right)\\ s.t.\left(u_{1},\ldots ,u_{N}\right)\in S,\left(u_{1},\ldots ,u_{N}\right)\geq \left(d_{1},\ldots ,d_{N}\right) \end{array}$$

The NB solution can be calculated as the vector that maximizes the product of the player’s utility gains relative to a fixed threat point. The threat point represents the maximum number of vehicles that could be accumulated per lane (i.e., the maximum measurable queue length). The objective is to minimize and equalize the queue lengths across the different phases. Hence, the negative queue length is used as the utility of each strategy considering a negative threat point. In other words, the reward (*u*) is defined to be the negative of the estimated queue length ($Q_{P}$), i.e., $u=-Q_{P}$, and we substitute (*d*) with a negative number. Consequently, the objective function can be rewritten as follows:(6)$$\begin{array}{c} \;\;\;\max_{\left(Q_{P1},\ldots ,Q_{PN}\right)}\prod_{i=1}^{N}\left(d_{i}-Q_{Pi}\right)\\ s.t.\left(Q_{P1},\ldots ,Q_{PN}\right)\in S,\left(Q_{P1},\ldots ,Q_{PN}\right)\leq \left(d_{1},\ldots ,d_{N}\right) \end{array}$$

The block diagram for the DNB controller is shown in Figure 3, where the predefined threat point values are an input to the controller (i.e., the maximum queue size that each player can accommodate). $Q_{outl}$ are generally measured at the approach stop bar, whereas $Q_{inl}$ are measured at a distance from the stop bar equal to the threat point divided by the approach jam density (i.e., the maximum length of the queue assuming all vehicles are stopped).

The flows $Q_{inl}$ and $Q_{outl}$ can be measured using stationary sensors (e.g., loop detectors or through video image processor (VIP) detection obtained from CCTV cameras). The queue length estimates can be obtained using CCTV cameras or via GPS-equipped vehicles that communicate with the the traffic signal controller. As such, the proposed controller is technology agnostic.

### 2.3. DNB Controller for Multi-Intersections

This section presents the DNB controller formulation for a network composed of multiple signalized intersections. For illustration purposes only, we formulate the problem considering three signalized intersections, as shown in Table 2. It should be noted, however that the algorithm can operate on a network of any number of signalized intersections.

Assume we have three signalized intersections ($I_{1},\;I_{2},\;I_{3}$), each traffic signal has three phases ($Ph_{1},\;Ph_{2},\;Ph_{3}$), where each phase is modeled as a player in a game resulting in a total of nine players where $I_{1}$ has three players ($P_{1},\;P_{2},\;P_{3}$), $I_{2}$ has three players ($P_{4},\;P_{5},\;P_{6}$), and $I_{3}$ has three players ($P_{7},\;P_{8},\;P_{9}$). Each traffic signal has three possible actions (*A*), where one phase displays a green indication (**G**) while the others display a red indication (**R**), as illustrated in Table 2.

Consequently, for the three signalized network illustrated in Table 2, there are 27 possible scenarios (action permutations) as shown in Table 3. The optimum overall network performance (NB optimum, Equation (Equation 6)) can be computed from Table 3.

Referring to Table 2, and assuming that the first traffic signal ($I_{1}$) has action ($A_{12}$) that optimizes its performance, traffic signal ($I_{2}$) has action ($A_{21}$) that optimizes its performance, and traffic signal ($I_{3}$) has action ($A_{33}$) that optimizes its performance. Consequently, searching in Table 3 for the Nash optimum combination yields scenario 12. This implies that in order to achieve the Nash optimum network performance, it is sufficient to search for the actions that optimize the operations of each signalized intersection. This can be described using the NB optimization problem shown in the following equations.
(7)$$\begin{array}{c} \;\;\;\;\max_{\left(u_{1},\ldots ,u_{9}\right)}\prod_{i=1}^{9}\left(u_{i}-d_{i}\right)\\ \;=\max_{\left(u_{1},\ldots ,u_{9}\right)}\left[\underset{I_{1}}{\underset{︸}{\prod_{i=1}^{3}\left(u_{i}-d_{i}\right)}}\;\;\underset{I_{2}}{\underset{︸}{\prod_{i=4}^{6}\left(u_{i}-d_{i}\right)}}\;\;\underset{I_{3}}{\underset{︸}{\prod_{i=7}^{9}\left(u_{i}-d_{i}\right)}}\right]\\ \;=\underset{I_{1}}{\underset{︸}{\max_{\left(u_{1},\ldots ,u_{3}\right)}\prod_{i=1}^{3}\left(u_{i}-d_{i}\right)}}\;\;\underset{I_{2}}{\underset{︸}{\max_{\left(u_{4},\ldots ,u_{6}\right)}\prod_{i=4}^{6}\left(u_{i}-d_{i}\right)}}\;\;\underset{I_{3}}{\underset{︸}{\max_{\left(u_{7},\ldots ,u_{9}\right)}\prod_{i=7}^{9}\left(u_{i}-d_{i}\right)}} \end{array}$$

The network-wide Nash optimum solution is obtained by maintaining the Nash optimum solution at each signalized intersection. As such, while the proposed NB controller is decentralized (i.e., DNB), it still produces the network-wide Nash-optimum control strategy relying solely on edge computing. The Nash optimum should not be mistaken for the system-optimum solution, where the system optimum might sacrifice the performance of one or more traffic signals to achieve optimum network-wide performance. It should be noted that obtaining the system-optimum solution is impossible given the scale and level of interactions of the various network-wide traffic signal controllers. The DNB controller, thus, provides a scalable and resilient controller that circumvents the problems inherent in complex centralized systems with minimum sacrifices to network-wide performance.

Note that a single traffic signal cannot be decomposed (i.e., optimize each decision variable independently), as the utilities of the players within the same traffic signal are dependent on each other. Specifically, if one player displays a green indication by default the other players have to display a red indication given that this would result in conflicting movements being discharged simultaneously. Alternatively, each traffic signal controller operates independently. Consequently, decomposition is invalid within a traffic signal but valid between traffic signals, as players within a traffic signal compete for the same resource, namely green time.

## 3. Blacksburg Town Experiments

This section presents the experimental setup and the results of a testing of the proposed system in the town of Blacksburg, Virginia, illustrated in Figure 4. The simulations were conducted using the morning peak hour (7–8 a.m.) traffic demand. The town of Blacksburg has 38 signalized intersections, 549 stop signs, 30 yield signs, and 1844 links. The minimum free-flow speed on the network was 30(km/h), and the maximum free-flow speed on the network was 105(km/h). The minimum link length was 50m while the maximum link length was 2932m. The jam density was set at 160(veh/km/lane). The traffic signal phasing scheme used in the study was the same as those in the field. These varied between 2 to 4 phases.

### 3.1. Blacksburg Experimental Setup

The time-dependent static O-D demand matrices were generated every 15 min using the QueensOD software [49,50,51]. QueensOD estimates the most likely O-D matrix that is as close structurally as a seed matrix while at the same time minimizing the error between the estimated and field observed link flow counts. The time-dependent static O-Ds were then used to compute a dynamic O-D matrix using procedures described in [52]. The final peak-hour dynamic O-D matrix consisted of 23,260 vehicular trips. Vehicles were loaded for one hour, while the simulation continued until all vehicles cleared the network to ensure that the same number of vehicles were used in comparing the performance of the various traffic signal control algorithms.

The performance of the DNB controller was evaluated by comparing its performance to that of the PSC and PSCO controllers. The network-wide average of each of the following measures of effectiveness (MOEs) was calculated to assess the DNB controller’s performance: travel time, total delay, stopped delay, queue length, fuel consumption, and emission levels. The INTEGRATION microscopic traffic assignment and simulation software was used to model the network, shown in Figure 4. Three experiments were conducted on the BB network, as discussed in the following sections.

### 3.2. BB Experimental Results: 1

In this experiment the performance of the DNB controller was compared to the PSC and PSCO controllers. The threat point (*d*) values per lane for the DNB controller were assigned based on the link’s lengths (*L*), the link’s free-flow speeds ($U_{f}$), and the updating time intervals (Δt), using the following formula; *d*=min[N(L/2), $N(U_{f}\times$Δt)], where N(L/2) represents the number of vehicles that could be accumulated up to the half length of the link, and $N(U_{f}\times$Δt) represents the maximum number of stopped vehicles that could be stored in the distance ($U_{f}\times$Δt). Using this distance allowed vehicles to proceed through the intersection in a minimal time without stopping if there was no queue ahead of them. A distance of L/2 was used instead of *L* to get a better estimate of the queue length for each movement because drivers typically moved to their desired lanes as they got closer to the signalized intersection, and to avoid being fully queued (i.e., players will accept a fully occupied (queued) link).

The average MOE values over the entire simulation for the PSC, PSCO, and DNB control scenarios are summarized in Table 4. In addition, Table 4 shows the percent improvement in MOEs using the proposed DNB controller over the PSC and PSCO controllers. The improvement (%) is calculated as:(8)$$\mathrm{Improvement}\left(\%\right)=\frac{\mathrm{MOE}(\mathrm{PSC}/\mathrm{PSCO})-\mathrm{MOE}(\mathrm{DNB})}{\mathrm{MOE}(\mathrm{PSC}/\mathrm{PSCO})}\times 100$$

The simulation results demonstrated a significant reduction in the average travel time of 5.25%, a reduction in the average total delay of 16.5%, and a reduction in the average stopped delay of 40.3% over the PSC controller. In addition, the results indicated significant reduction in the average travel time of 6.5%, a reduction in the average total delay of 19.8%, and a reduction in the average stopped delay of 52.7% over the PSCO controller. These results show that the proposed DNB controller outperforms both the PSC and PSCO controllers.

### 3.3. BB Experimental Results: 2

This section presents a potential solution to better estimate the queue length considering the driver’s lane changing behavior close to the intersections. A suggested phasing scheme, shown in Figure 5b, where all vehicles on the link discharge in a single phase, might provide a better estimate of the queue length per phase over the currently implemented phase scheme shown in Figure 5a, where each link discharges in two phases. Two simulations were conducted using the DNB controller to evaluate the effectiveness of the two phasing scheme on the MOEs, where the threat point per lane was assigned using the following formula: *d*=min[N(L/2), $N(U_{f}\times \Delta t)$]. The simulation results using the two schemes (Figure 5) are shown in Table 5.

The simulation results demonstrate that the suggested phasing scheme does not improve the network performance.

### 3.4. BB Experimental Results: 3

This section presents the effect of reducing the number of vehicles that can be accumulated in a lane on the network’s performance. The minimum free-flow speed on the network was 30(km/h), and the maximum free-flow speed on the network was 105(km/h), with updating time intervals of 10s. Assigning the detector locations to be the min(L/2, $U_{f}\times \Delta t$), the detectors could be located for long links between 84 m (i.e., 13 veh/lane) to 292 m (i.e., 47 veh/lane). Employing the free-flow speed to determine the threat point (*d* = min[N(L/2), $N(U_{f}\times \Delta t)$]) is a good choice for low traffic demand, as vehicles are not required to stop at the intersection); however, for high traffic demand, long links can accommodate long queues, which causes delays for the vehicles on that link. Hence, reducing the number of vehicles that can accumulate in a lane might enhance the network’s performance. To examine the effectiveness of changing the maximum number of vehicles that could be accumulated per lane on the MOEs, a sensitivity analysis was conducted, as shown in Figure 6, with *d* = min[N(L/2), NV], where NV presents the maximum number of vehicles that can be stored in a lane; this number ranges between 6 to 32 vehicles.

Analysis of the results in Figure 6 demonstrated that better performance using the DNB controller could be achieved if the threat points are assigned as a minimum of 12 veh/lane and the number of vehicles that could be accumulated in L/2, (*d*=min[N(L/2), 12]).

Table 6 shows the average MOEs values over the entire simulation time and the percent improvement in MOEs using the proposed DNB controller over PSC and PSCO controllers. Simulation results indicate significant reduction in the average total delay of 19.38%, a reduction in the average stopped delay of 51.18%, a reduction in the average travel time of 6.162%, a reduction in the average number of stops of 8.39%, a reduction in the average fuel consumption of 3.89%, and a reduction in the emission levels of 3.84% over the PSC controller. The results show that the proposed DNB approach outperforms both the PSC and PSCO controllers.

To further investigate the achieved improvements using the DNB controller, it was taken into consideration that the network has 459 stop signs and 30 yield signs, which might conceal the full degree of improvement achieved using the DNB controller on the signalized intersection. Accordingly, we investigated the percent improvement in MOEs using the DNB controller over the PSC controller over only the links that were directly associated with intersections. Table 7 shows the percent improvement in MOEs using the DNB controller over the PSC controller on the 38 intersections.

Table 7 demonstrates an improvement in the travel time on the intersections between 6% to 52%, an improvement in the queue length on the intersections between 8% to 60%, and an improvement in the number of stops on the intersections between 8% to 80%. In addition, Table 7 demonstrates an overall reduction in the average travel time of 23.63%, in the average queued vehicles of 37.66%, in the average number of stops of 23.58%, in the average fuel consumption of 10.44%, in the average ${\mathrm{CO}}_{2}$ emitted of 9.84%, and in the average ${\mathrm{NO}}_{X}$ emitted of 5.4% over the PSC controller. These results revealed that the DNB controller performs significantly better than the PSC controller.

## 4. Downtown Los Angeles Experiments

This section describes the experimental setup and the experimental results of large scale studies in downtown Los Angeles, California comprised of 457 signalized intersections.

### 4.1. Los Angeles Experimental Setup

These experiments were large scale studies of a network in downtown Los Angeles (LA), California, including the most congested downtown area, as shown in Figure 7a. The INTEGRATION microscopic traffic assignment and simulation software was used to model the network, as shown in Figure 7b.

Simulations were conducted using the morning peak hour (7–8 a.m.) traffic demand that was calibrated in a previous effort [53]. The downtown LA network has 457 signalized intersections, 285 stop signs, 23 yield signs, and 3556 links. The origin-destination (O-D) demand matrices were generated, as described earlier, using a combination of the QueensOD software, to generate time-dependent static O-D demands, and then converting these static O-D demands to a dynamic O-D demand. The resulting O-D consisted of a total of 143,957 vehicle trips. Vehicles were loaded for the one-hour period and the simulation continued until all vehicles cleared the network to ensure that all comparisons were made for the same number of vehicles.

The traffic signal phasing schemes varied from 2 to 6 phases, reflecting the field implemented traffic signal settings in downtown LA. The minimum free-flow speed on the network was 15(km/h), and the maximum free-flow speed on the network was 120(km/h). The minimum link length on the network was 50m, and the maximum link length on the network was 4400m. The jam density of the various network links was set equal to 180(veh/km/lane).

The DNB controller was compared to the PSC controller to evaluate their relative performance. The average of each of the following measures of effectiveness (MOEs) was calculated to assess the performance of the DNB controller: travel time, total delay, stopped delay, queue length, fuel consumption, and emission levels.

### 4.2. LA Experimental Results: 1

In this experiment, the performance of the DNB controller was compared to that of the PSC controller using the full traffic demand in the morning peak hour. The threat point per lane for the DNB controller was assigned as the minimum of 12 veh/lane and the number of vehicles that could be accumulated on L/2 (i.e., *d* = min[N(L/2), 12]) based on the sensitivity analysis shown in Figure 8.

The average MOE values over the entire simulation for the PSC and DNB controllers are shown in Table 8. In addition, Table 8 shows the percent improvement in MOEs using the proposed DNB controller relative to the PSC controller. The simulation results demonstrate a significant reduction in the average travel time of 7.89%, a reduction in the total delay of 14.55%, a reduction in the average stopped delay of 25.18%, a reduction in the average number of vehicle stops of 12.4%, a reduction in the average fuel consumption of 4.0%, and a reduction in ${\mathrm{CO}}_{2}$ emission levels of 4.25%, relative to the PSC controller. Analysis of the results demonstrated that the proposed DNB controller outperforms current state-of-the-art de-centralized traffic signal controllers.

The improvements produced by the DNB controller, only at the signalized intersections, were further analyzed. Accordingly, we investigated the percent improvement in MOEs using the DNB controller over the PSC controller over only the links that were directly associated with signalized intersections.

Table 9 demonstrates a reduction in the average travel time of 35.16%, a reduction in the average queued vehicles of 54.67%, a reduction in the average number of stops of 44.03%, a reduction in the average fuel consumption of 9.97%, a reduction in the ${\mathrm{CO}}_{2}$ emissions of 9.92%, and a reduction in the ${\mathrm{NO}}_{X}$ emissions of 11.78% relative to the PSC controller. These results revealed that the DNB controller has significantly better performance potential than the PSC controller.

### 4.3. LA Experimental Results: 2

A simulation was conducted for lower levels of traffic congestion by scaling the demand down by 90% (i.e., 10% of the peak demand) to investigate the performance potential using the DNB controller. Table 10 shows a reduction in the average travel time of 7.1%, a reduction in the average total delay of 36.79%, a reduction in the average stopped delay of 90.26%, a reduction in the average number of vehicle stops of 34.66%, a reduction in the average fuel consumption of 4.8%, and a reduction in ${\mathrm{CO}}_{2}$ emission levels of 4.79%, relative to the PSC controller.

Once more, to further investigate the achieved improvements using the DNB controller, we investigated the improvement in MOEs over only the links that were directly associated with signalized intersections, as shown in Table 11. Table 11 demonstrates a reduction in the average travel time of 19.19%, a reduction in the average queued vehicles of 49.84%, a reduction in the average number of stops of 53.71%, a reduction in the average fuel consumption of 54.16%, a reduction in the average ${\mathrm{CO}}_{2}$ emitted of 16.09%, and a reduction in the average ${\mathrm{NO}}_{X}$ emitted of 25.94% over PSC controller.

These results demonstrate that the DNB controller performed significantly better than the PSC controller in both congested and uncongested conditions, however, produced more savings as the traffic demand decreased.

The results show that the DNB controller yielded significant improvements in the average values of all MOEs, demonstrating improved system efficiency.

## 5. Summary & Conclusions

The research presented in this paper develops and evaluates a Nash bargaining de-centralized flexible phasing cycle-free traffic signal controller (DNB controller) on large-scale networks. The controller was implemented and tested in the INTEGRATION microscopic traffic assignment and simulation software. The performance of the DNB controller was compared to a decentralized phase split and cycle length optimization controller based on the HCM procedures (PSC) and a fully-coordinated adaptive phase split, cycle length and offset optimization controller (PSCO), in the town of Blacksburg, Virginia and in downtown Los Angeles, California.

Several simulations were conducted on the Blacksburg network using different threat point values and phasing schemes to determine their effect on the controller’s performance. The results show significant reductions in the network-wide average travel time of 6.1% and 7.3%, a reduction in the average total delay of 19.3% and 22.6%, a reduction in the stopped delay of 51% and 61%, and a reduction in ${\mathrm{CO}}_{2}$ emission levels of 3.8% and 3.7%, over the PSC and PSCO controllers, respectively. In addition, the results show significant reductions on the intersection approach average travel time of 23.6%, a reduction in the average queue length of 37.6%, a reduction in the average number of vehicle stops of 23.6%, a reduction in the fuel consumption of 9.8%, a reduction in the ${\mathrm{CO}}_{2}$ emissions of 10.4%, and a reduction in ${\mathrm{NO}}_{X}$ emissions of 5.4%.

In addition, the DNB controller’s performance was tested in downtown Los Angeles, California, and compared to the performance of the de-centralized PSC controller. The results show significant improvements in various network-wide measures of performance. Specifically, a reduction in the average travel time of 8%, a reduction in the average total delay of 14.5%, a reduction in the stopped delay of 25.1%, a reduction in the average number of vehicle stops of 12.4%, and a reduction in ${\mathrm{CO}}_{2}$ emissions of 4.25%, over the PSC controller. Moreover, the results show significant improvements in the signalized intersection operations with a reduction in the average travel time of 35.1%, a reduction in the average queue length of 54.7%, a reduction in the average number of vehicle stops of 44%, a reduction in the fuel consumption and ${\mathrm{CO}}_{2}$ emissions of 10%, and a reduction in ${\mathrm{NO}}_{X}$ emissions of 11.7%. Furthermore, simulations conducted for lower traffic demand levels showed significant network-wide improvements with a reduction in the average total delay of 36.7%, a reduction in the stopped delay of 90.2%, and a reduction in the average number of stops of 35% over the PSC controller. As these results indicate, the DNB controller can generate major performance improvements at lower demands. The results demonstrate significant potential benefits of using the proposed controller over other state-of-the-art centralized and de-centralized controllers on large scale networks.

In summary, a novel traffic signal controller is developed that offers a number of unique features. First, the controller adapts signal timings dynamically to changing traffic conditions without using historical data, which tends to be inaccurate, resulting in inefficient traffic signal plans. Second, the developed controller is de-centralized, which increases both the scalability and robustness of the system, to avoid the problems inherent with complex centralized communication. Decentralized systems are often inexpensive to establish and operate, as there is no essential need for a reliable and direct communication network between a central computer and the local controllers in the field. Third, the controller, while de-centralized, does not sacrifice in system-wide performance and computes the network-wide Nash optimum solution. Finally, the controller is designed to operate with current traffic signal controllers. This controller should increase the traffic handling capacity of roads, and reduce unnecessary stop-and-go vehicular movement, which will reduce fuel consumption and, accordingly, air pollution.

## Figures and Tables

**Figure 1 sensors-19-02282-f001:**
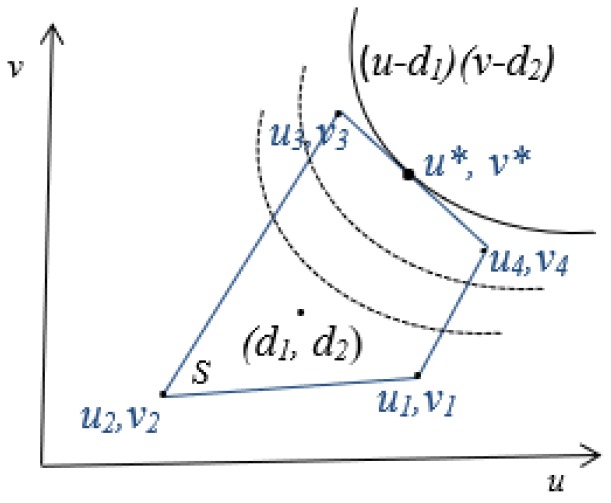
Utility region.

**Figure 2 sensors-19-02282-f002:**
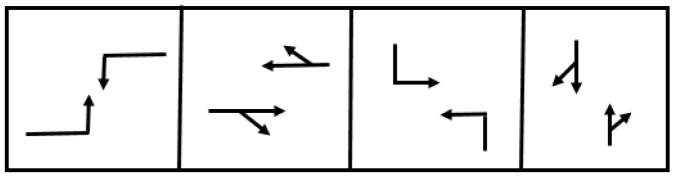
Phasing scheme.

**Figure 3 sensors-19-02282-f003:**
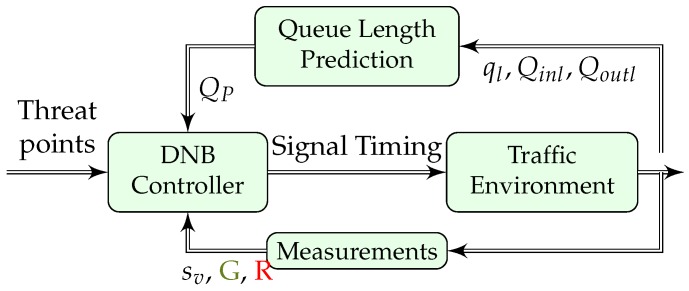
System block diagram.

**Figure 4 sensors-19-02282-f004:**
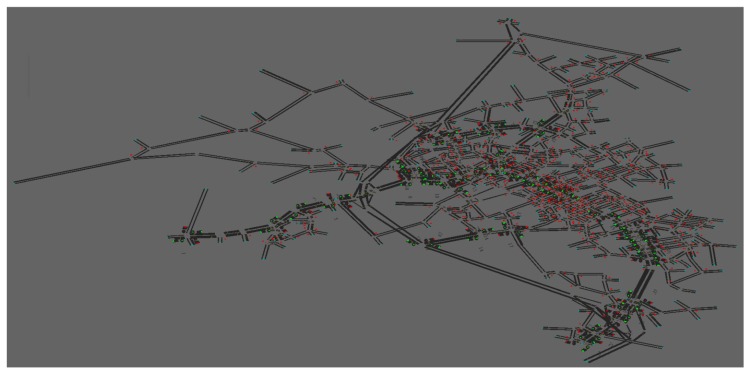
Blacksburg network.

**Figure 5 sensors-19-02282-f005:**
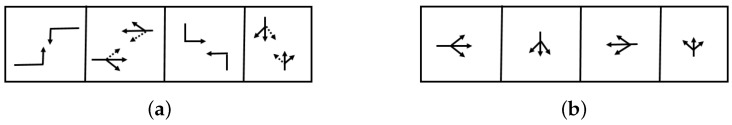
Four phasing scheme. (**a**) Implemented phasing scheme. (**b**) Suggested phasing scheme.

**Figure 6 sensors-19-02282-f006:**
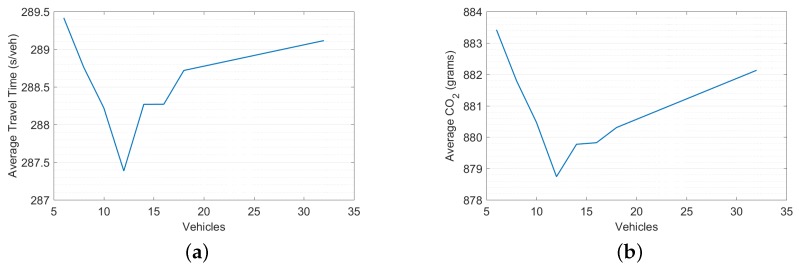
Sensitivity analysis. (**a**) Average travel time. (**b**) Average ${\mathrm{CO}}_{2}$.

**Figure 7 sensors-19-02282-f007:**
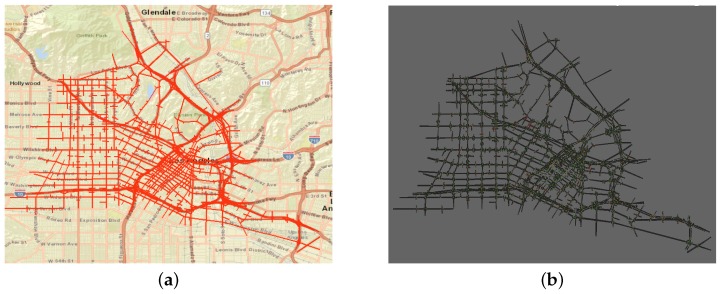
Downtown Los Angeles network. (**a**) LA, Google maps. (**b**) LA, INTEGRATION.

**Figure 8 sensors-19-02282-f008:**
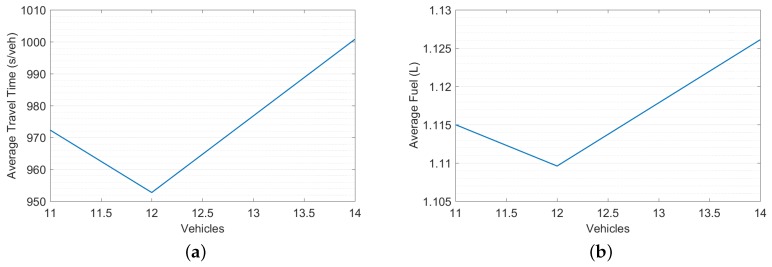
LA Sensitivity Analysis. (**a**) Average Travel Time. (**b**) Average Fuel Consumption.

**Table 1 sensors-19-02282-t001:** Two players matrix game.

		$P_{2}$	
		$A_{1}$	$A_{2}$
$P_{1}$	$A_{1}$	$u_{1},v_{1}$	$u_{2},v_{2}$
	$A_{2}$	$u_{3},v_{3}$	$u_{4},v_{4}$

**Table 2 sensors-19-02282-t002:** Multi-player matrix game.

Intersection		First Intersection (*I*_1_)			Second Intersection (*I*_2_)			Third Intersection (*I*_3_)		
	Player	Ph1 (*P*_1_)	Ph2 (*P*_2_)	Ph3 (*P*_3_)	Ph1 (*P*_4_)	Ph2 (*P*_5_)	Ph3 (*P*_6_)	Ph1 (*P*_7_)	Ph2 (*P*_8_)	Ph3 (*P*_9_)
Action										
First		$\underset{A_{11}}{\underset{︸}{\begin{array}{ccc} G & R & R \end{array}}}$			$\underset{A_{21}}{\underset{︸}{\begin{array}{ccc} G & R & R \end{array}}}$			$\underset{A_{31}}{\underset{︸}{\begin{array}{ccc} G & R & R \end{array}}}$		
Second		$\underset{A_{12}}{\underset{︸}{\begin{array}{ccc} R & G & R \end{array}}}$			$\underset{A_{22}}{\underset{︸}{\begin{array}{ccc} R & G & R \end{array}}}$			$\underset{A_{32}}{\underset{︸}{\begin{array}{ccc} R & G & R \end{array}}}$		
Third		$\underset{A_{13}}{\underset{︸}{\begin{array}{ccc} R & R & G \end{array}}}$			$\underset{A_{23}}{\underset{︸}{\begin{array}{ccc} R & R & G \end{array}}}$			$\underset{A_{33}}{\underset{︸}{\begin{array}{ccc} R & R & G \end{array}}}$		

**Table 3 sensors-19-02282-t003:** All possible Network Actions (Permutations).

Scenario #	Network Action
1	$A_{11}\;\;A_{21}\;\;A_{31}$
2	$A_{11}\;\;A_{21}\;\;A_{32}$
3	$A_{11}\;\;A_{21}\;\;A_{33}$
4	$A_{11}\;\;A_{22}\;\;A_{31}$
5	$A_{11}\;\;A_{22}\;\;A_{32}$
6	$A_{11}\;\;A_{22}\;\;A_{33}$
7	$A_{11}\;\;A_{23}\;\;A_{31}$
8	$A_{11}\;\;A_{23}\;\;A_{32}$
9	$A_{11}\;\;A_{23}\;\;A_{33}$
10	$A_{12}\;\;A_{21}\;\;A_{31}$
11	$A_{12}\;\;A_{21}\;\;A_{32}$
12	$A_{12}\;\;A_{21}\;\;A_{33}$
13	$A_{12}\;\;A_{22}\;\;A_{31}$
14	$A_{12}\;\;A_{22}\;\;A_{32}$
15	$A_{12}\;\;A_{22}\;\;A_{33}$
16	$A_{12}\;\;A_{23}\;\;A_{31}$
17	$A_{12}\;\;A_{23}\;\;A_{32}$
18	$A_{12}\;\;A_{23}\;\;A_{33}$
19	$A_{13}\;\;A_{21}\;\;A_{31}$
20	$A_{13}\;\;A_{21}\;\;A_{32}$
21	$A_{13}\;\;A_{21}\;\;A_{33}$
22	$A_{13}\;\;A_{22}\;\;A_{31}$
23	$A_{13}\;\;A_{22}\;\;A_{32}$
24	$A_{13}\;\;A_{22}\;\;A_{33}$
25	$A_{13}\;\;A_{23}\;\;A_{31}$
26	$A_{13}\;\;A_{23}\;\;A_{32}$
27	$A_{13}\;\;A_{23}\;\;A_{33}$

**Table 4 sensors-19-02282-t004:** Average measures of effectiveness (MOEs) and (%) improvement for game-theoretic framework (DNB) over phase split and cycle length controller (PSC) and phase split-cycle length and offset optimization controller (PSCO) controllers.

	System	PSC	PSCO	DNB
MOE				
Average Total Delay (s/veh)		96.234	100.197	80.323
Improvement %		16.534	19.823	
Average Stopped Delay (s/veh)		20.285	25.649	12.1074
Improvement %		40.314	52.7962	
Average Travel time (s)		306.254	310.225	290.175
Improvement %		5.250	6.463	
Average Number of Stops		4.662	4.5899	4.281
Improvement %		8.18	6.734	
Average Fuel (L)		0.4142	0.4129	0.40
Improvement %		3.38	3.07	
Average ${\mathrm{CO}}_{2}$ Emissions (g)		913.833	912.495	883.127
Improvement %		3.36	3.22	

**Table 5 sensors-19-02282-t005:** MOEs using two different phasing schemes.

	System	DNB (Field Scheme)	DNB (Modified Scheme)	Imp. (%)
MOE				
Average Total Delay (s/veh)		80.323	94.712	−17.913
Average Stopped Delay (s/veh)		12.107	24.381	−101.374
Average Travel Time (s)		290.175	302.425	−4.222
Average Number of Stops		4.281	4.417	−3.177
Average Fuel (L)		0.40	0.41	−2.274
Average ${\mathrm{CO}}_{2}$ Emissions (g)		883.127	902.277	−2.168

**Table 6 sensors-19-02282-t006:** Average MOEs and (%) improvement using DNB over the PSC and PSCO controllers.

	System	PSC	PSCO	DNB
MOE				
Average Total Delay (s/veh)		96.234	100.197	77.577
Improvement %		19.3871	22.575	
Average Stopped Delay (s/veh)		20.285	25.649	9.903
Improvement %		51.182	61.391	
Average Travel Time (s)		306.254	310.225	287.384
Improvement %		6.162	7.362	
Average Number of Stops		4.662	4.5899	4.271
Improvement %		8.393	6.95	
Average Fuel (L)		0.4142	0.4129	0.3981
Improvement %		3.887	3.584	
Average CO ${}_{2}$ (grams)		913.833	912.495	878.739
Improvement %		3.84	3.7	

**Table 7 sensors-19-02282-t007:** Intersections (%) improvement of MOEs using DNB over PSC controller.

	MOEs	Travel Time	Queue	Num. of Stops	${\mathrm{CO}}_{2}$	Fuel	${\mathrm{NO}}_{X}$
Int. #							
1		6.153	22.015	24.311	2.645	2.566	0.161
2		16.409	26.801	21.184	7.706	7.710	5.859
3		8.485	18.233	32.777	6.034	6.450	9.040
4		31.114	52.874	39.564	8.166	6.595	8.756
5		22.230	53.875	52.914	9.355	8.962	3.309
6		23.176	34.435	14.240	11.594	10.716	4.751
7		8.967	15.881	17.832	3.889	3.597	2.271
8		24.057	41.868	16.114	13.753	13.480	9.162
9		40.709	56.267	29.850	25.253	24.654	13.842
10		13.395	26.346	41.436	8.634	8.653	9.772
11		17.628	26.340	11.802	9.014	8.353	1.352
12		7.642	7.968	32.650	3.481	3.373	3.476
13		19.414	37.909	20.915	8.991	8.745	3.758
14		28.503	35.499	25.359	7.854	6.617	8.147
15		23.870	39.630	34.584	12.553	12.272	6.166
16		27.552	59.095	41.876	15.109	14.785	8.836
17		42.001	60.000	56.974	16.896	14.827	12.842
18		26.258	49.883	32.723	14.491	13.414	5.703
19		19.676	36.533	21.104	4.963	4.253	4.976
20		52.237	76.083	63.088	32.966	31.762	20.159
21		34.822	50.159	46.265	21.568	21.385	18.268
22		38.267	59.396	37.466	27.628	27.284	26.528
23		17.193	30.863	16.272	7.595	6.922	5.258
24		34.669	43.997	11.269	14.632	13.342	3.239
25		23.480	44.588	57.381	5.760	4.502	0.085
26		18.029	26.028	30.503	4.017	2.478	0.750
27		28.129	36.340	8.565	16.769	16.194	14.480
28		14.530	35.046	11.902	9.459	9.846	11.611
29		13.131	19.115	9.603	5.347	4.985	1.142
30		23.632	47.382	23.224	19.330	19.409	24.772
31		32.761	55.701	80.381	18.004	17.273	19.333
32		34.761	53.070	35.456	26.641	27.045	29.311
33		35.984	48.472	15.256	20.348	19.563	11.668
34		16.679	32.676	30.335	11.273	11.151	11.757
35		18.012	28.950	21.575	18.241	18.672	26.116
36		22.588	46.509	34.331	7.676	7.028	2.465
37		29.307	46.502	31.486	7.399	6.678	1.081
38		14.317	14.552	8.061	4.669	4.168	1.143
**Overall (%)**		23.633	37.666	23.586	10.444	9.842	5.390

**Table 8 sensors-19-02282-t008:** Average MOEs and the (%) improvement using DNB controller over PSC controller (100% Demand).

	System	PSC	DNB	DNB Imp. (%)
MOE				
Average Total Delay (s/veh)		557.463	476.346	14.55
Average Stopped Delay (s/veh)		256.766	192.116	25.178
Average Travel Time (s)		1034.27	952.732	7.89
Average Number of Stops		7.406	6.487	12.4
Average Fuel (L)		1.155	1.109	4.0
Average ${\mathrm{CO}}_{2}$ (grams)		2482.13	2376.59	4.25

**Table 9 sensors-19-02282-t009:** Average (%) improvements of MOEs using DNB controller over PSC controller (100% Demand), over the links that are directly associated with intersections.

	MOEs	Travel Time	Queue	Num. of Stops	${\mathrm{CO}}_{2}$	Fuel	${\mathrm{NO}}_{X}$
Int. #							
Overall 457 Int. (%)		35.156	54.66	44.031	9.966	9.919	11.774

**Table 10 sensors-19-02282-t010:** Average MOEs and the (%) improvement using DNB over PSC controller (10% Demand).

	System	PSC	DNB	DNB Imp. (%)
MOE				
Average Total Delay (s/veh)		84.938	53.689	36.79
Average Stopped Delay (s/veh)		19.971	1.9451	90.261
Average Travel Time (s)		450.114	418.177	7.1
Average Number of Stops		4.475	2.924	34.66
Average Fuel (L)		0.846	0.805	4.8
Average ${\mathrm{CO}}_{2}$ (grams)		1830.27	1742.53	4.79

**Table 11 sensors-19-02282-t011:** Average (%) improvements of MOEs using DNB over PSC controller (10% Demand) over the links directly associated with intersections.

	MOEs	Travel Time	Queue	Num. of Stops	${\mathrm{CO}}_{2}$	Fuel	${\mathrm{NO}}_{X}$
Int. #							
Overall 457 Int. (%)		19.186	49.844	53.708	54.158	16.085	25.939

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
