# Peer review of "A Novel Decentralized Game-Theoretic Adaptive Traffic Signal Controller: Large-Scale Testing"

_sensors, 2019, doi:10.3390/s19102282_

Round 1
Reviewer 1 Report
This paper proposes a new light controller using the Nash bargaining concept. It was found that by finding the best solution for each intersection would improve the systems` decision while taking into account real time data.
Please find some comments below:
Page 3: 32-38. Is there any advantage for using centralized systems? I understand that the authors want to justify de-centralization. However, centralized systems should have at least one advantage.
Page 5: 15-18: Could you please elaborate more this idea? Does this relate to the ability of a vehicle to pass the light before the light switches color? If so i can understand that it will form into the queue. However, once in the queue, how can it gain speed? What controls the vehicle speed?
I am as well thinking about the distance. How far from the intersection are you considering to set this threshold speed? Is this important or am i going too far?
Otherwise, what do you mean by vehicle speed?
Page 5: Eq (3). It might seem evident that you are comparing against a previous instant, but please explain the (t-1) in this equation.
Page 5: Eq 4. Do you assume each line for each phase? or only the lines for the interested player (phase). As well Qinl and Qoutl is a parameter PER line or do you assume the same for each line?
Page 5: Eq 5. It would be a good idea to show your "new" objective function.
Page 5: 34. How do you select these threat points? (I see that you define how later on). It would be useful to include this in the methodology section. Explain all the variables please.
Page 6: 11. Figure 3 and text are too close together.
Page 6: 20. What is "Best Scenario"? Please define/remind it.
Page 6: 26. So, the problem can be reduced to maximize each player? I want to make sure I understood correctly.
Page 8. 14-15. Was not this supposed to be measured with a minimal time?
Page 8: 7. Please clarify that this experiment does not take into account the driver behaviour. If a vehicle is "assigned" a line, it will stay on that line.
Page 9: 6. Please explain the purpose of this test.
Page 9: 8. Please define " detectors locations"
References:
Some Years are bold and others are not. Is there a reason why?
Avoid using "we" please.
Page 3: line 39
Page 4: line 42
Page 5: line 31.
Page 6: line 20.
Thank you
Author Response
{\color{dblue} \item Page 3: 32-38. Is there any advantage for using centralized systems? I understand that the authors want to justify de-centralization. However, centralized systems should have at least one advantage.}
\\
Centralized systems require a reliable and direct communication network between a central computer and the local controllers. The main advantage of these systems is that they allow for traffic signal coordination. We added this to the paper (see page 3).
%-----------------------------(1.2)--------------------------------
{\color{dblue} \item Page 5: 15-18: Could you please elaborate more this idea? Does this relate to the ability of a vehicle to pass the light before the light switches color? If so i can understand that it will form into the queue. However, once in the queue, how can it gain speed? What controls the vehicle speed?
\\I am as well thinking about the distance. How far from the intersection are you considering to set this threshold speed? Is this important or am i going too far?
\\Otherwise, what do you mean by vehicle speed?}
\\
The INTEGRATION software is a microscopic traffic simulation model that traces individual vehicle movements every deci-second. Driver characteristics such as reaction times, acceleration and deceleration levels, desired speeds, and lane-changing behavior are examples of stochastic variables that are modeled in INTEGRATION.
The threshold speed is fixed and assigned to the entire network (chosen to be equal to the typical pedestrian speed, $s^{Th}$= $\rm{4.5~(km/h)}$). We continuously check the vehicle speeds when they are within the threat distance from the approach stop bar. If the vehicle ($v$) speed ($s_v^t$) is less than the threshold speed ($s^{Th}$) at time ($t$), the vehicle is assigned to the queue, and the current queue length associated with the corresponding lane ($l$) is updated. Once the vehicle's speed exceeds ($s^{Th}$) the queue length is updated (i.e., shortened by the number of vehicles leaving the queue). We have amended the text, see page 5. We also added more description of the INTEGRATION software and its various components relative to this effort.
%-----------------------------(1.3)--------------------------------
{\color{dblue} \item Page 5: Eq (3). It might seem evident that you are comparing against a previous instant, but please explain the (t-1) in this equation.}
\\
S(t-1) refers to the speed in the previous time step. We added a clarification to the paper.
%-----------------------------(1.4)--------------------------------
{\color{dblue} \item Page 5: Eq 4. Do you assume each line for each phase? or only the lines for the interested player (phase). As well Qinl and Qoutl is a parameter PER line or do you assume the same for each line?}
\\
The utilities for each player (phase) in the game is defined as the estimated sum of the queue lengths associated with each phase (that includes all movement in that phase).
The flow ratios $Q_{inl}$ and $Q_{outl}$ are for the movements in each phase.
%-----------------------------(1.5)--------------------------------
{\color{dblue} \item Page 5: Eq 5. It would be a good idea to show your "new" objective function.}
\\
Thanks for this comment, we have amended the text, and added a new equation for the objective function, see Equation 6 on page 6.
%-----------------------------(1.6)--------------------------------
{\color{dblue} \item Page 5: 34. How do you select these threat points? (I see that you define how later on). It would be useful to include this in the methodology section. Explain all the variables please.}
\\
The threat point represents the maximum number of vehicles that could be accumulated per lane (i.e., the maximum measurable queue length), see page 6 line 2, and page 6 line 10.
%-----------------------------(1.7)--------------------------------
{\color{dblue} \item Page 6: 11. Figure 3 and text are too close together. }
\\
We fixed that.
%-----------------------------(1.8)--------------------------------
{\color{dblue} \item Page 6: 20. What is "Best Scenario"? Please define/remind it.}
\\
The best scenario is the solution to the NB optimization problem, equation 6, see page 7 line 3.
%-----------------------------(1.9)--------------------------------
{\color{dblue} \item Page 6: 26. So, the problem can be reduced to maximize each player? I want to make sure I understood correctly.}
\\
Each phase in an intersection is considered a player and only one player can hold the green indication at any instant in time. Consequently, to achieve the maximum network performance it is sufficient to search for the actions that will maximize the intersection performance (each intersection has more than one player) separately.
%-----------------------------(1.10)--------------------------------
{\color{dblue} \item Page 8. 14-15. Was not this supposed to be measured with a minimal time? }
\\
We are not quite sure what the question is, however we added the following "Using the distance of $ \rm{U_f} \times$ ${\Delta t}$ allowed detected vehicles to pass the intersection in a minimal time without stopping if there was no queue in front of them...", see page 8 line 34.
%-----------------------------(1.11)--------------------------------
{\color{dblue} \item Page 8: 7. Please clarify that this experiment does not take into account the driver behaviour. If a vehicle is "assigned" a line, it will stay on that line.}
\\
The INTEGRATION software is a microscopic traffic simulation model that traces individual vehicle movements every deci-second. Driver characteristics such as reaction times, acceleration and deceleration rates, desired speeds, car following model, and lane-changing behavior are examples of stochastic variables that are incorporated in INTEGRATION. This experiment takes into account the driver behaviour ( where drivers usually moved to the appropriate lanes as they got closer to the signalized intersection), see see page 8 line 36.
%-----------------------------(1.12)--------------------------------
{\color{dblue} \item Page 9: 6. Please explain the purpose of this test.}
\\
This section presents the effect of reducing the number of vehicles that can be accumulated in a lane on the network’s performance. , see see page 10 line 2.
%-----------------------------(1.13)--------------------------------
{\color{dblue} \item Page 9: 8. Please define " detectors locations"}
\\
Output detectors are generally located at the downstream end of the links, whereas input detector are located at distances from the downstream end of the links equal to threat points over jam densities, see page 6 line 11.
%-----------------------------(1.14)--------------------------------
{\color{dblue} \item References:
Some Years are bold and others are not. Is there a reason why?}
\\
We noticed that the journals papers' year are bold while other are non-bold, it might be this journal formatting style.
%-----------------------------(1.15)--------------------------------
{\color{dblue} \item Avoid using "we" please. Page 3: line 39, Page 4: line 42, Page 5: line 31, Page 6: line 20.}
\\
This comment has been fixed, see page 3 line 43, see page 5 line 4 see page 6 line 5, and see page 7 line 3.
Reviewer 2 Report
This manuscript presents decentralized game-theoretic adaptive traffic signal controller: large-scale simulation. The subject is very interesting. There are some comments and questions:
1. There is a problem in Figure 3. Its contain block named 'DNB Controller' but the caption is 'DNB controller block diagram.' This Figure gives the impression that it is ill-conceived. Arrows should be named so that to know what data they represent, or what measurement is needed for the controller.
2. We can see in table 4 ‘Improvement %’ while in the text there is talk about ‘Reduction.’ For clear-at-glance, you can add equation formally defines this parameter.
3. Mistakes/ shortcomings:
- on page 4, line 3: 'Los Angles',
- on page 12, line 17: 'more, To’.
- In the whole manuscript, for nitrogen oxides, there is literally ‘NOX,’ ‘Nox,’ and ‘NOx.’ Please normalized to the last one.
- Table 10 contain in the caption ‘(0.1% Demand)’, but on page 12, line 12 it is ‘(i.e., 10% of the peak demand)’.
- There is a lack of information about 'Author Contributions' at the end of the manuscript.
- There is a lack of DOI in references. Please add.
- In the Tables, there are numerical results presented in resolution ‘.001’. It would be easier to read results with lower resolution. Is there the sense to show of the thousandth part of % at all?
4. Does DNB controller work in LA or BB nowadays? I'm guessing it's not. What effort should be made to introduce into the real world the DNB controller? Where should modifications be done, in software only as well as in hardware? What kind of modification LA or BB needs that introduce DNB controller? Are there advanced enough the traffic measurement system there?
5. Is it any technical reasons, which can block introduce the DNB controller?
6. Will the DNB controller works better if there is information about vehicles class, not only length of the Queue? Can accurate measurements of the parameters of vehicles standing in the queue be somehow useful?
Author Response
\item {\bf \underline{{Reviewer 2 Comments:}}}
\begin{enumerate}[label*=\arabic*.]
%-----------------------------(2.1)--------------------------------
{\color{dblue} \item There is a problem in Figure 3. Its contain block named 'DNB Controller' but the caption is 'DNB controller block diagram.' This Figure gives the impression that it is ill-conceived. Arrows should be named so that to know what data they represent, or what measurement is needed for the controller.}
\\
The caption has been changed to 'System block diagram', and arrows have been labeled.
%-----------------------------(2.2)--------------------------------
{\color{dblue} \item We can see in table 4 ‘Improvement \%’ while in the text there is talk about ‘Reduction.’ For clear-at-glance, you can add equation formally defines this parameter.}
\\
Equation (8) has been added.
%-----------------------------(2.3)--------------------------------
{\color{dblue} \item Mistakes/ shortcomings:
\\- on page 4, line 3: 'Los Angles', {\color{black}\checkmark}
\\- on page 12, line 17: 'more, To’. {\color{black}\checkmark}
\\- In the whole manuscript, for nitrogen oxides, there is literally ‘NOX,’ ‘Nox,’ and ‘NOx.’ Please normalized to the last one. {\color{black}\checkmark}
\\- Table 10 contain in the caption ‘(0.1\% Demand)’, but on page 12, line 12 it is ‘(i.e., 10\% of the peak demand)’. {\color{black}\checkmark}
\\- There is a lack of 'Author Contributions' at the end of the manuscript. {\color{black}\checkmark}
\\- There is a lack of DOI in references. Please add. {\color{black}\checkmark}
\\- In the Tables, there are numerical results presented in resolution ‘.001’. It would be easier to read results with lower resolution. Is there the sense to show of the thousandth part of \% at all?} {\color{black}\checkmark}
\\
These comments have been addressed.
%-----------------------------(2.4)--------------------------------
{\color{dblue} \item Does DNB controller work in LA or BB nowadays? I'm guessing it's not. What effort should be made to introduce into the real world the DNB controller? Where should modifications be done, in software only as well as in hardware? What kind of modification LA or BB needs that introduce DNB controller? Are there advanced enough the traffic measurement system there?}
\\
No the DNB is not been field employed in BB nor LA. We only implemented them in a virtual simulation environment. The DNB controller lends itself conveniently to field implementation because the control decision from the DNB controller can be translated into standard NTCIP commands (green hold/force-off as the key ones) to influence the signals at the target intersections. No hardware modifications are needed at the traffic signals.
%-----------------------------(2.5)--------------------------------
{\color{dblue} \item Is it any technical reasons, which can block introduce the DNB controller?}
\\
There is no technical reason that can block the DNB controller, it is technology agnostic.
%-----------------------------(2.6)--------------------------------
{\color{dblue} \item Will the DNB controller works better if there is information about vehicles class, not only length of the Queue? Can accurate measurements of the parameters of vehicles standing in the queue be somehow useful?}
\\
We are working on introducing the vehicle class to the DNB controller, the preliminary results look promising. This would allow for transit traffic signal priority.

Reviewer 3 Report
The reviewer thinks the methodology is not correct. In Equation (6), how can you decompose the problem from a higher dimension to a low dimension just by separating the decision variables by intersections? It is a constrained optimization problem. Also the intersections are correlated to each other, on traffic flow, they are not independent. Just take Equation 1 as a simple example. If the decomposition is valid, then equation 1 can be written as:
max_u(u-d1) * max_v(v-d2)
Then you will get (u4, v3) as your optimal solution, not u*v* in figure 1..
Actually it is a very interesting topic that using decentralized mechanism to control a large network of traffic signals..
Author Response
\item {\bf \underline{{Reviewer 3 Comments:}}}
\begin{enumerate}[label*=\arabic*.]
%-----------------------------(3.1)--------------------------------
{\color{dblue} \item The reviewer thinks the methodology is not correct. In Equation (6), how can you decompose the problem from a higher dimension to a low dimension just by separating the decision variables by intersections? It is a constrained optimization problem. Also the intersections are correlated to each other, on traffic flow, they are not independent. Just take Equation 1 as a simple example. If the decomposition is valid, then equation 1 can be written as: \rm{$max_u(u-d1) * max_v(v-d2) $}
Then you will get (u4, v3) as your optimal solution, not u*v* in figure 1. Actually it is a very interesting topic that using decentralized mechanism to control a large network of traffic signals.}
\\
To achieve the maximum network performance it is sufficient to search for the actions that maximize intersection performance individually, the decomposition is valid down to the level of intersection, as players within the intersection cannot all have a green indication at the same time (i.e. they share same resource (green time)), however multi-intersections do not compete for the same green indications and thus are independent.

Round 2
Reviewer 3 Report
The reviewer is not convinced by the authors explainations. Although actions of each intersection are independent, but the constraints are not. For example, the arrival flow of an intersection is related to the traffic signal (actions) of the upstream intersection. In addition, the authors didn't answer the reviewer's question regarding equation 1 and figure 1: If Equation 1 is decomposed the same way as equation 7, you will get a total different solution. Optimizing each decision variable (dimension) independently doesn't result in a global optimal solution.
Author Response
We would like to thank the anonymous reviewer for their valuable comments regarding our manuscript. We are pleased to submit a revised version of our manuscript entitled \emph{A Novel Decentralized Game-theoretic Adaptive Traffic Signal Controller: Large-scale Testing} to the Journal of Sensors. Every effort was made to address the comments raised by the reviewer, as summarized below.
%%% ----------------------- Reviewer3
\item {\bf \underline{{Reviewer 3 Comments:}}}
\begin{enumerate}[label*=\arabic*.]
%%%%%%%%%%%-------------------------------------------(3.1)
{\color{dblue} \item The reviewer is not convinced by the authors explanations. Although actions of each intersection are independent, but the constraints are not. For example, the arrival flow of an intersection is related to the traffic signal (actions) of the upstream intersection. }
\\
With regards to this comment, we need to clarify three points:
\begin{enumerate}[label=(\alph*)]
\item The only constraints we include in our formulation relate to the threat points. These represent the maximum number of queued vehicles that we are willing to accept for each phase. These are not affected by the actions of other traffic signals in the network. Consequently, our constraints are independent of the actions of other signalized intersections.
\item The reviewer is correct in that the actions of uptstream traffic signals affect the arrivals at downstream signals. This interaction is captured through the measured inflows to each approach. One could correctly argue that the inflows are outcomes of control at upstream intersections and thus one could add an additional player to capture the offset between traffic signals. This would render the controller a centralized controller and then we would have to maintain the same cycle length across all traffic signals to ensure traffic signal coordination. We did compare our algorithm to the REALTRAN controller (PSCO controller), which attempts to achieve system-wide coordination. In our paper we demonstrated that our decentralized approach outperforms the centralized PSCO approach. That is because we have a trade-off between local optima and coordination. If we go for a centralized approach we have to sacrifice the performance of the individual intersections to maintain a constant cycle length. Alternatively, if we go with a decentralized approach we have each traffic signal operate at its optimum but fail to optimize the coordination between traffic signals (i.e. the offsets). In our paper we demonstrate that it is more efficient if we run each traffic signal freely and sacrifice the coordination.
\item Our proof that the network optimum is achieved through the optimization of each controller separately is because we are computing the Nash optimum solution and not the system-optimum solution. This distinction is very important. This is very similar to the traffic assignment problem where you have a user-optimum versus a system-optimum solution (Wardrop's first and second principal). Using this analogy, the DNB controller computes the user optimum solution, where the user in our case is the traffic signal controller (i.e. each traffic signal controller is optimized to achieve a user equilibrium). The system-optimum solution is impossible to derive given the complexity of the problem and the various traffic signal interactions. However, as we demonstrate in the paper, the Nash optimum solution outperforms all state-of-the-art approaches that we compared with. We have modified the paper to make sure that we explicitly state that it is the Nash optimum solution that we are deriving.
\end{enumerate}
{\color{dblue} \item In addition, the authors didn't answer the reviewer's question regarding equation 1 and figure 1: If Equation 1 is decomposed the same way as equation 7, you will get a total different solution. Optimizing each decision variable (dimension) independently doesn't result in a global optimal solution.}
\\
We agree with the reviewer that Eq. 1 cannot be decomposed. It should be noted that Section 2.1 (including Figure 1 and Eq. 1) describes the NB approach in general.
Following, is a description of why we can decompose Eq. 7 (multiple traffic signals) and cannot decompose Eq. 1 (single traffic signal). For illustration, the game matrix (Table \ref{tab:MG}) and the optimization problem (Eq. \ref{eq:NEW}) are shown below for an isolated intersection having two phases (two players). Each player (phase) has two possible actions ({\color{DGreen}G} or {\color{red}R} ), where one phase displays a green indication ({\color{DGreen}G}) and the other displays a red indication ({\color{red}R}). Both players cannot display green or red indications simultaneously, as illustrated in Table~\ref{tab:MG}.
%% Table O_D
\begin {table}[h!]
\caption {Two players (phases) matrix game for an isolated intersection}
\label{tab:MG}
\centering
\setlength\tabcolsep{12pt} % default value: 6pt
\begin{tabular}[l]{ccc|c}
& &\multicolumn{2}{c}{{$P_2$}} \\ \cline{3-4}
& & \color{dblue}${\color{red}R}$&\color{DGreen}G\\
\specialrule{1pt}{0pt}{0pt}
\multirow{2}{*}{\begin{rotate}{90}$P_1$\end{rotate}}&\color{DGreen}G& \multicolumn{1}{!{\VRule[1pt]}c| }{${u_{1\color{DGreen}G},u_{2\color{red}R}}$ }&${-}$\\ \cline{2-4}
&\color{dblue}${\color{red}R}$& \multicolumn{1}{!{\VRule[1pt]}c| }{${-}$} &${u_{1\color{red}R},u_{2\color{DGreen}G}}$ \\
\specialrule{1pt}{0pt}{0pt}
\end{tabular}
\end{table}
The solution for the two-phase NB problem can be formulated as:
\begin{equation}
\begin{aligned}
\label{eq:NEW}
&\quad \quad \underset{{u_1,u_2}}{\rm{max}}{~(u_1-d_1)(u_2-d_2)}\\
& i.e., {\rm{max}}\big[~(u_{1\color{DGreen}G}-d_1)(u_{2\color{red}R}-d_2),~(u_{1\color{red}R}-d_1)(u_{2\color{DGreen}G}-d_2)\big]
\end{aligned}
\end{equation}
Note, Equation 1 (for a single traffic signal) cannot be decomposed (i.e., optimize each decision variable (dimension) independently), as the utilities of the players ($u_1,u_2$) are dependent on each other. In other words, we cannot calculate $u_2$ without knowing $u_1$. Specifically, if we calculate $u_1$ for player 1 assuming a green indication (i.e., ${u_{1\color{DGreen}G}}$) then, we have to calculate $u_2$ for player 2 considering a red indication (i.e., $u_{2\color{red}R}$), and vise versa.
In summary, the two players (phases) cannot display a green indication simultaneously given that this would violate the whole premise of a traffic signal. Hence, Optimizing each decision variable (dimension) independently at the traffic signal level (Eq.1) is not possible given that it would produce unrealistic traffic signal timings.
On the other hand, across traffic signals, the traffic signal phases are independent of each other with each controller operating independently. Consequently, decomposition is invalid within a traffic signal but valid between traffic signals, as players within a traffic signal have to operate within safety constraints.
Note, in Eq. 7 we are not optimizing each decision variable independently ($u_1..u_9$), we break the equation down to the traffic signal level, i.e., traffic signal 1 ($u_1-u_3$), traffic signal 2 ($u_4-u_6$), traffic signal 3 ($u_7-u_9$). We cannot optimize $u_1-u_3$ individually as they belong to the same traffic signal, however traffic signal 2 ($u_4-u_6$) can be optimized independent of traffic signal 1 ($u_1-u_3$) as they do not compete for the same traffic signal indications, please refer to section 2.3.
In summary, decomposition is not allowed within a traffic signal (Eq.1) as players are constrained by safety requirements (separating conflicting movements), however, decomposition is allowed between traffic signals (Eq.7) where such constraints do not exist.

Round 3
Reviewer 3 Report
Thanks for the clarification, I have no more comments regarding methodology.